# WORD SEQUENCE PREDICTION FOR AMHARIC LANGUAGE

September 2019

## Abstract

The significance of computers, handheld devices and educational applications are not deniable in the modern world of today. The interest of applying AI and Machine Learning algorithms in this technologies is growing. Texts are entered to these devices using word processing programs as well as other techniques. Word prediction is one of the techniques and assistive technology for people with disabilities. The main purpose of word prediction is to facilitate data entry and help dyslexic people in reducing writing errors. Dyslexia is a learning problem some kids and few adults have. Dyslexia makes it tough to read and spell. For developing countries such as Ethiopia this kind of problems are neglected and the language spoken within the country are under resourced. Therefore this work has a major contribution to those who have this type of disability in their own language. Prediction can also help education by facilitating data entry. Building prediction model with better speed, grammatically correct suggestions and less search space is the main focus of the research. Amharic is used by a large number of populations, however no significant work is done on the topic of word sequence prediction. In this study, Amharic word sequence prediction model is developed with statistical methods using Hidden Markov Model by incorporating detailed parts of speech tag, some morphological features and user profiling or adaptation. Evaluation of the model is performed using developed prototype and keystroke savings (KSS) as a metrics. According to our experiment, prediction results using a bi-gram with morphological features and detailed Parts of Speech tag model has higher KSS and performed better compared those without Parts of Speech tag. Therefore, statistical approach with detailed POS, morphological features like gender, number, and person with suggested root or stem words using voice, tense, aspect, affixes statistical information and grammatical agreement rules of the language has quite good potential on word sequence Prediction for Amharic language.

## 1  Introduction

This research deals with designing word sequence prediction model in Amharic language. It is a language that is spoken in eastern Africa (Wimsatt Wynn,

2011). One of the needs for Amharic word sequence prediction for mobile use and other digital device is in order to facilitate data entry and communication in our language especially for those who has Dyslexia. Word sequence prediction is a challenging task for inflected languages (Gustavii Pettersson, 2003; Seyyed Assi, 2005). These kinds of languages are morphologically rich and have enormous word forms, which is word can have different forms. As Amharic language is highly inflected language and morphologically rich it shares the problem (Tessema, 2014).This problem makes word prediction system much more difficult and results poor performance. Due to this reason, storing all forms in a dictionary won't solve the problem as in English and other less inflected languages. Hence considering machine learning algorithms that could help the predictor to suggest the next word with POS based prediction and the stated morphological features should be used.

## 1.1 background of study

Previous researches used dictionary approach with no consideration of context information. Hence storing all forms of words in dictionary for inflected languages such as Amharic language has been less effective. The main goal of this thesis is to implement Amharic word prediction model that works with better Prediction speed and with narrowed search space as much as possible. We introduced two models; tags and words and linear interpolation that use parts of speech tag information in addition to word n-grams in order to maximize the likelihood of syntactic appropriateness of the suggestions. We believe the results found reflect this.

# 2 Methodology

The study followed Design Science Research Methodology (DSRM). Since DSRM includes approaches, techniques, tools, algorithms and evaluation mechanisms in the process, we followed statistical approach with statistical language modeling using Machine learning algorithms and built Amharic prediction model based on information from Parts of Speech tagger. The statistics included in the systems varies from single word frequencies to parts-of-speech tag n-grams. That means it included the statistics of Word frequencies, Word sequence frequencies, Parts-of-speech sequence frequencies and other important morphological information. Later on, the system was evaluated using Keystroke Savings. (Trnka Mccoy, 2008) Linux mint was used as the main Operating System during the frame work design. We used 31 tag sets that has been developed by linguistics professionals from different domains, python programming language and its libraries for both the language model and the predictor module. Another Tool that was used is the SRILIM (The SRI language modeling toolkit) which was used to generate unigram, bigram and trigram count as an input for the language model. SRILIM is a toolkit that is used to build and apply Statistical language modeling (Levy, 2015).

# 3 Conclusion

This study presents Amharic word sequence prediction model using the statistical approach. We described a combined statistical and lexical word prediction system for handling inflected languages by making use of POS tags with morphological features to build the language model using Hidden Markov Model, TNT tagger. We developed Amharic language models of bigram and trigram for the training purpose. We obtained 29We concluded that employing syntactic information in the form of Parts-of-Speech (POS) and morphological features with n-grams promises more effective predictions. We also can conclude that data quantity, performance of POS tagger and data quality highly affects the keystroke savings. According to our evaluation, better Keystroke saving (KSS) is achieved when using bi-gram model than the tri-gram models and higher n-grams. This is because in statistical modeling with higher n-grams requires huge amount of data. We believe the results obtained were effective in reflecting better speed, correctness of suggestions (grammatical), and search space since these are the basic issues in word sequence prediction and in assistive technology.

## 3.1 Limitation

This research has limitation of data quantity and quality. We have tried our best to collect data sets and building such models needs much more data in statistical approach. The quality of POS Tagger has quite negative effect on the predictor module We have a plan of extending this work using neural network algorithms like recurrent neural network and CNN with a better POS model. This algorithms can overcome the data problem we face because they can work effectively with small data. We also would like to adopt this work for other local languages that has many speakers in Ethiopia.

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
