# OpenReview forum: "WORD SEQUENCE PREDICTION FOR AMHARIC LANGUAGE"
_ICLR.cc/2020/Conference — Reject_

### Official Review · AnonReviewer3 · 2019-10-14
**Official Blind Review #3**

**Rating:** 1

**Review:**

I would not like to sound offensive, but this paper is clearly below the standards of the conference, and outside any academic orthodoxy for the matter:
- It is only 3 pages long including references, and does not even follow the conference template.
- It has only 3 sections ("introduction", "methodology" and "conclusions") and 2 subsections ("background of study" and "limitation"), and all of them are only one paragraph long.
- The work done is not adequately described, so it is not possible to say much about it, but it seems clear that there is no novelty nor sufficient rigor in it. The problem tackled is defined as "word prediction", which seems to be some form of language modeling. The proposed method combines HMMs with n-grams and morphological and POS features, which are all well established and should be considered more of a (nowadays outdated) baseline. There is no proper evaluation: the experimental settings are not described, and only one number is reported, with nothing to compare to.

To the authors: Please do not feel discouraged by my review. Your motivation (helping people with dyslexia in Ethiopia) is certainly laudable, but neither the work carried out nor its presentation meets the standards of our research community. I would suggest that you check some of the accepted papers in last year's conference to get a sense of what is expected. I assume that you are new to the field. Don't feel the need to publish your own research from the first day, and take your time to become familiar with the field and study the basic concepts. It takes time, but we all had to go through it :)

**Experience Assessment:**

I have published in this field for several years.

**Review Assessment: Checking Correctness Of Derivations And Theory:**

N/A

**Review Assessment: Checking Correctness Of Experiments:**

N/A

**Review Assessment: Thoroughness In Paper Reading:**

I read the paper thoroughly.

---

### Official Review · AnonReviewer1 · 2019-10-22
**Official Blind Review #1**

**Rating:** 1

**Review:**

Summary:

This paper proposes to predict word sequences for Amharic language-- a language spoken in Eastern Africa. It proposes to use HMMs with POS tags and morphological features to perform this prediction task.


The paper is just 3 pages, contains 1 paragraph of methodology, and no experiments section. It is clearly a very early stage work and not in the scope of ICLR. This paper should have been desk-rejected as it needs more work before it is fit for publication. There is lot of work on word sequence prediction and HMMs are no longer the state-of-the-art. The authors should consider looking at RNN-based methods such as LSTMs for this task.

**Experience Assessment:**

I have published one or two papers in this area.

**Review Assessment: Checking Correctness Of Derivations And Theory:**

N/A

**Review Assessment: Checking Correctness Of Experiments:**

N/A

**Review Assessment: Thoroughness In Paper Reading:**

I read the paper thoroughly.

---

### Official Review · AnonReviewer2 · 2019-11-03
**Official Blind Review #2**

**Rating:** 1

**Review:**

*What is this paper about?*

The authors propose a method to incorporate POS tags into a language model to improve its performance in Amharic language.

Short review:

The authors tackle an interesting task which deserves more attention. Nonetheless, they do not fully describe their models or results with enough detail, so it is hard to evaluate this work.


Contributions:

This work tackles a relevant problem that seriously impacts speakers of low resource languages.

*What strengths does this paper have?*

It tackles and interesting problem.

*What weaknesses does this paper have?*

The authors do not present their models in enough details so that the reader fully understands it.
They also only gloss over the results, not presenting them in any concrete form, stating: “We believe the results obtained were effective in reflecting bet-ter speed, correctness of suggestions (grammatical), and search space since these are the basic issues in word sequence prediction and in assistive technology”. This referred results are not shown in the manuscript though.


*Detailed comments:*

The paper does not use the official conference template. It is also very short, not going in details about the used techniques. Finally, references are not correctly formatted.

Section 1.


**Experience Assessment:**

I have read many papers in this area.

**Review Assessment: Checking Correctness Of Derivations And Theory:**

I assessed the sensibility of the derivations and theory.

**Review Assessment: Checking Correctness Of Experiments:**

I assessed the sensibility of the experiments.

**Review Assessment: Thoroughness In Paper Reading:**

I read the paper thoroughly.

---

### Decision · Program_Chairs · 2019-12-19

**Decision:**

Reject

**Comment:**

This paper presents a language model for Amharic using HMMs and incorporating POS tags. The paper is very short and lacks essential parts such as describing the exact model and the experimental design and results. The reviewers all rejected this paper, and there was no author rebuttal. This paper is clearly not appropriate for publication at ICLR.